# Sparse nonnegative deconvolution for compressive calcium imaging: algorithms and phase transitions

**Eftychios A. Pnevmatikakis and Liam Paninski**
Department of Statistics, Center for Theoretical Neuroscience
Grossman Center for the Statistics of Mind, Columbia University, New York, NY
{eftychios, liam}@stat.columbia.edu

## Abstract

We propose a compressed sensing (CS) calcium imaging framework for monitoring large neuronal populations, where we image randomized projections of the spatial calcium concentration at each timestep, instead of measuring the concentration at individual locations. We develop scalable nonnegative deconvolution methods for extracting the neuronal spike time series from such observations. We also address the problem of demixing the spatial locations of the neurons using rank-penalized matrix factorization methods. By exploiting the sparsity of neural spiking we demonstrate that the number of measurements needed per timestep is significantly smaller than the total number of neurons, a result that can potentially enable imaging of larger populations at considerably faster rates compared to traditional raster-scanning techniques. Unlike traditional CS setups, our problem involves a block-diagonal sensing matrix and a non-orthogonal sparse basis that spans multiple timesteps. We provide tight approximations to the number of measurements needed for perfect deconvolution for certain classes of spiking processes, and show that this number undergoes a "phase transition," which we characterize using modern tools relating conic geometry to compressed sensing.

## 1 Introduction

Calcium imaging methods have revolutionized data acquisition in experimental neuroscience; we can now record from large neural populations to study the structure and function of neural circuits (see e.g. Ahrens et al. (2013)), or from multiple locations on a dendritic tree to examine the detailed computations performed at a subcellular level (see e.g. Branco et al. (2010)). Traditional calcium imaging techniques involve a raster-scanning protocol where at each cycle/timestep the microscope scans the image in a voxel-by-voxel fashion, or some other predetermined pattern, e.g. through random access multiphoton (RAMP) microscopy (Reddy et al., 2008), and thus the number of measurements per timestep is equal to the number of voxels of interest. Although this protocol produces "eye-interpretable" measurements, it introduces a tradeoff between the size of the imaged field and the imaging frame rate; very large neural populations can be imaged only with a relatively low temporal resolution.

This unfavorable situation can potentially be overcome by noticing that many acquired measurements are redundant; voxels can be "void" in the sense that no neurons are located there, and active voxels at nearby locations or timesteps will be highly correlated. Moreover, neural activity is typically sparse; most neurons do not spike at every timestep. During recent years, imaging practitioners have developed specialized techniques to leverage this redundancy. For example, Nikolenko et al. (2008) describe a microscope that uses a spatial light modulator and allows for the simultaneous imaging of different (predefined) image regions. More broadly, the advent of compressed sensing (CS) has found many applications in imaging such as MRI (Lustig et al., 2007), hyperspectral imaging (Gehm et al., 2007), sub-diffraction microscopy (Rust et al., 2006) and ghost imaging (Katz et al., 2009),

with available hardware implementations (see e.g. Duarte et al. (2008)). Recently, Studer et al. (2012) presented a fluorescence microscope based on the CS framework, where each measurement is obtained by projection of the whole image on a random pattern. This framework can lead to significant undersampling ratios for biological fluorescence imaging.

In this paper we propose the application of the imaging framework of Studer et al. (2012) to the case of neural population calcium imaging to address the problem of imaging large neural populations with high temporal resolution. The basic idea is to not measure the calcium at each location individually, but rather to take a smaller number of "mixed" measurements (based on randomized projections of the data). Then we use convex optimization methods that exploit the sparse structure in the data in order to simultaneously demix the information from the randomized projection observations and deconvolve the effect of the slow calcium indicator to recover the spikes. Our results indicate that the number of required randomized measurements scales merely with the number of expected spikes rather than the ambient dimension of the signal (number of voxels/neurons), allowing for the fast monitoring of large neural populations. We also address the problem of estimating the (potentially overlapping) spatial locations of the imaged neurons and demixing these locations using methods for nuclear norm minimization and nonnegative matrix factorization. Our methods scale linearly with the experiment length and are largely parallelizable, ensuring computational tractability. Our results indicate that calcium imaging can be potentially scaled up to considerably larger neuron populations and higher imaging rates by moving to compressive signal acquisition.

In the traditional static compressive imaging paradigm the sensing matrix is dense; every observation comes from the projection of all the image voxels to a random vector/matrix. Moreover, the underlying image can be either directly sparse (most of the voxels are zero) or sparse in some orthogonal basis (e.g. Fourier, or wavelet). In our case the sensing matrix has a block-diagonal form (we can only observe the activity at one specific time in each measurement) and the sparse basis (which corresponds to the inverse of the matrix implementing the convolution of the spikes from the calcium indicator) is non-orthogonal and spans multiple timelags. We analyze the effect of these distinctive features in Sec. 3 in a noiseless setting. We show that as the number of measurements increases, the probability of successful recovery undergoes a phase transition, and study the resulting phase transition curve (PTC), i.e., the number of measurements per timestep required for accurate deconvolution as a function of the number of spikes. Our analysis uses recent results that connect CS with conic geometry through the "statistical dimension" (SD) of descent cones (Amelunxen et al., 2013). We demonstrate that in many cases of interest, the SD provides a very good estimate of the PTC.

## 2   Model description and approximate maximum-a-posteriori inference

See e.g. Vogelstein et al. (2010) for background on statistical models for calcium imaging data. Here we assume that at every timestep an image or light field (either two- or three-dimensional) is observed for a duration of $T$ timesteps. Each observed field contains a total number of $d$ voxels and can be vectorized in a single column vector. Thus all the activity can be described by $d \times T$ matrix $F$. Now assume that the field contains a total number of $N$ neurons, where $N$ is in general unknown. Each spike causes a rapid increase in the calcium concentration which then decays with a time constant that depends on the chemical properties of the calcium indicator. For each neuron $i$ we assume that the "calcium activity" $\boldsymbol{c}_i$ can be described as a stable autoregressive process $AR(1)$ process[1] that filters the neuron's spikes $s_i(t)$ according to the fast-rise slow-decay procedure described before:

$$c_i(t) = \gamma c_i(t-1) + s_i(t), \tag{1}$$

where $\gamma$ is the discrete time constant which satisfies $0 < \gamma < 1$ and can be approximated as $\gamma = 1 - \exp(-\Delta t/\tau)$, where $\Delta t$ is the length of each timestep and $\tau$ is the continuous time constant of the calcium indicator. In general we assume that each $s_i(t)$ is binary due to the small length of the timestep in the proposed compressive imaging setting, and we use an i.i.d. prior for each neuron $p(s_i(t) = 1) = \pi_i$.[2] Moreover, let $\boldsymbol{a}_i \in \mathbb{R}_+^d$ the (nonnegative) location vector for neuron $i$, and $\boldsymbol{b} \in \mathbb{R}_+^d$ the (nonnegative) vector of baseline concentration for all the voxels. The spatial calcium concentration profile at time $t$ can be described as

$$\boldsymbol{f}(t) = \sum\nolimits_{i=1}^{N} \boldsymbol{a}_i c_i(t) + \boldsymbol{b}. \tag{2}$$

In conventional raster-scanning experiments, at each timestep we observe a noisy version of the $d$-dimensional image $\boldsymbol{f}(t)$. Since $d$ is typically large, the acquisition of this vector can take a significant amount of time, leading to a lengthy timestep $\Delta t$ and low temporal resolution. Instead, we propose to observe the *projections* of $\boldsymbol{f}(t)$ onto a random matrix $B_t \in \mathbb{R}^{n \times d}$ (e.g. each entry of $B_t$ could be chosen as 0 or 1 with probability 0.5):

$$\boldsymbol{y}(t) = B_t \boldsymbol{f}(t) + \boldsymbol{\varepsilon}_t, \quad \boldsymbol{\varepsilon}_t \sim \mathcal{N}(\boldsymbol{0}, \Sigma_t), \tag{3}$$

where $\boldsymbol{\varepsilon}_t$ denotes measurement noise (Gaussian, with diagonal covariance $\Sigma_t$, for simplicity). If $n = \dim(\boldsymbol{y}(t))$ satisfies $n \ll d$, then $\boldsymbol{y}(t)$ represents a compression of $\boldsymbol{f}(t)$ that can potentially be obtained more quickly than the full $\boldsymbol{f}(t)$. Now if we can use statistical methods to recover $\boldsymbol{f}(t)$ (or equivalently the location $\boldsymbol{a}_i$ and spikes $\boldsymbol{s}_i$ of each neuron) from the compressed measurements $\boldsymbol{y}(t)$, the total imaging throughput will be increased by a factor proportional to the undersampling ratio $d/n$. Our assumption here is that the random projection matrices $B_t$ can be constructed quickly. Recent technological innovations have enabled this fast construction by using digital micromirror devices that enable spatial light modulation and can construct different excitation patterns with a high frequency (order of kHz). The total fluorescence can then be detected with a single photomultiplier tube. For more details we refer to Duarte et al. (2008); Nikolenko et al. (2008); Studer et al. (2012).

We discuss the statistical recovery problem next. For future reference, note that eqs. (1)-(3) can be written in matrix form as ($\text{vec}(\cdot)$ denotes the vectorizing operator)

$$
\begin{aligned}
S &= CG^T \\
F &= AC + \boldsymbol{b}\boldsymbol{1}_T^T \\
\text{vec}(Y) &= B\text{vec}(F) + \boldsymbol{\varepsilon},
\end{aligned}
\quad \text{with } G =
\begin{bmatrix}
1 & 0 & \dots & 0 \\
-\gamma & 1 & \dots & 0 \\
\vdots & \ddots & \ddots & \vdots \\
0 & \dots & -\gamma & 1
\end{bmatrix}, \; B = \text{blkdiag}\{B_1, \dots, B_T\}. \tag{4}
$$

## 2.1  Approximate MAP inference with an interior point method

For now we assume that $A$ is known. In general MAP inference of $S$ is difficult due to the discrete nature of $S$. Following Vogelstein et al. (2010) we relax $S$ to take continuous values in the interval $[0, 1]$ (remember that we assume binary spikes), and appropriately modify the prior for $s_i(t)$ to $\log p(s_i(t)) \propto -(\lambda_i s_i(t))1(0 \le s_i(t) \le 1)$, where $\lambda_i$ is chosen such that the relaxed prior has the same mean $\pi_i$. To exploit the banded structure of $G$ we seek the MAP estimate of $C$ (instead of $S$) by solving the following convex quadratic problem (we let $\bar{\boldsymbol{y}}(t) = \boldsymbol{y}(t) - B_t \boldsymbol{b}$)

$$
\begin{aligned}
&\underset{C}{\text{minimize}} && \sum_{t=1}^{T} \frac{1}{2}(\bar{\boldsymbol{y}}(t) - B_t A \boldsymbol{c}(t))^T \Sigma_t^{-1} (\bar{\boldsymbol{y}}(t) - B_t A \boldsymbol{c}(t)) - \log p(C) \\
&\text{subject to} && 0 \le CG^T \le 1, \boldsymbol{c}(1) \ge 0,
\end{aligned}
\tag{P-QP}
$$

Using the prior on $S$ and the relation $S = CG^T$, the log-prior of $C$ can be written as $\log p(C) \propto -\boldsymbol{\lambda}^T CG^T \boldsymbol{1}_T$. We can solve (P-QP) efficiently using an interior point method with a log-barrier (Vogelstein et al., 2010). The contribution of the likelihood term to the Hessian is a block-diagonal matrix, whereas the barrier-term will contribute a block-tridiagonal matrix where each non-zero block is diagonal. As a result the Newton search direction $-H^{-1}\nabla$ can be computed efficiently in $O(TN^3)$ time using a block version of standard forward-backward methods for tridiagonal systems of linear equations. We note that if $N$ is large this can be inefficient. In this case we can use an augmented Lagrangian method (Boyd et al., 2011) to derive a fully parallelizable first order method, with $O(TN)$ complexity per iteration. We refer to the supplementary material for additional details.

As a first example we consider a simple setup where all the parameters are assumed to be known. We consider $N = 50$ neurons observed over $T = 1000$ timesteps. We assume that $A, \boldsymbol{b}$ are known, with $A = I_N$ (corresponding to non-overlapping point neurons, with one neuron in each voxel) and $\boldsymbol{b} = \boldsymbol{0}$, respectively. This case of known point neurons can be thought as the compressive analog of RAMP microscopy where the neuron locations are predetermined and then imaged in a serial manner. (We treat the case of unknown and possibly overlapping neuron locations in section 2.2.) Each neuron was assumed to fire in an i.i.d. fashion with probability per timestep $p = 0.04$. Each measurement was obtained by projecting the spatial fluorescence vector at time $t$, $\boldsymbol{f}(t)$, onto a random matrix $B_t$. Each row of $B_t$ is taken as an i.i.d. normalized vector $2\boldsymbol{\beta}/\sqrt{N}$, where $\boldsymbol{\beta}$ has i.i.d. entries following a fair Bernoulli distribution. For each set of measurements we assume that $\Sigma_t = \sigma^2 I_n$, and

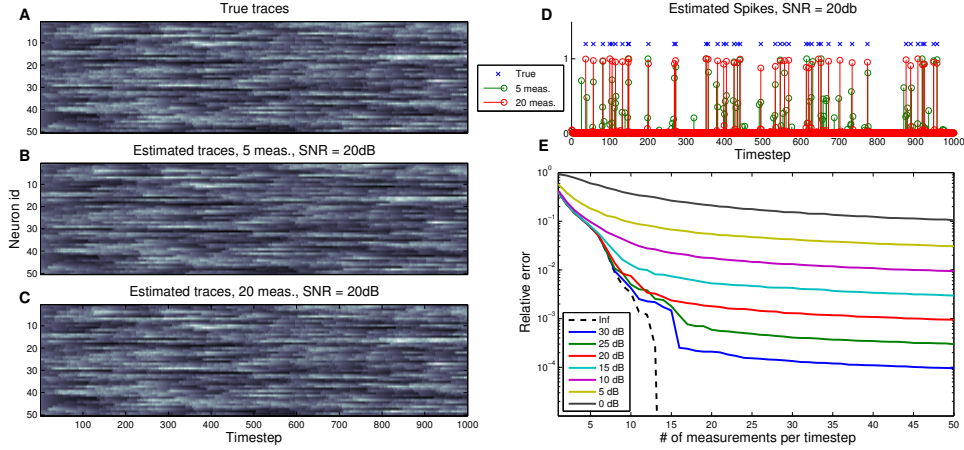

Figure 1: Performance of proposed algorithm under different noise levels. A: True traces, B: Estimated traces with $n = 5$ ($10\times$ undersampling), SNR = 20dB. C: Estimated traces with $n = 20$ ($2.5\times$ undersampling), SNR = 20dB. D: True and estimated spikes from the traces shown in panels B and C for a randomly selected neuron. E: Relative error between true and estimated traces for different number of measurements per timestep under different noise levels. The error decreases with the number of observations and the reconstruction is stable with respect to noise.

the signal-to-noise ratio (SNR) in dB is defined as $\text{SNR} = 10\log_{10}(\text{Var}[\boldsymbol{\beta}^T \boldsymbol{f}(t)]/N\sigma^2)$; a quick calculation reveals that $\text{SNR} = 10\log_{10}(p(1-p)/(1-\gamma^2)\sigma^2)$.

Fig. 1 examines the solution of (P-QP) when the number of measurements per timestep $n$ varied from 1 to $N$ and for 8 different SNR values $0, 5, \ldots, 30$ plus the noiseless case (SNR $= \infty$). Fig. 1A shows the noiseless traces for all the neurons and panels B and C show the reconstructed traces for SNR = 20dB and $n = 5, 20$ respectively. Fig. 1D shows the estimated spikes for these cases for a randomly picked neuron. For very small number of measurements ($n = 5$, i.e., $10\times$ undersampling) the inferred calcium traces (Fig. 1B) already closely resemble the true traces. However, the inferred MAP values of the spikes (computed by $S = CG^T$, essentially a differencing operation here) lie in the interior of $[0, 1]$, and the results are not directly interpretable at a high temporal resolution. As $n$ increases ($n = 20$, red) the estimated spikes lie very close to $\{0, 1\}$ and a simple thresholding procedure can recover the true spike times. In Fig. 1E the relative error between the estimated and true traces ($\|C - \hat{C}\|_F/\|C\|_F$, with $\|\cdot\|_F$ denoting the the Frobenius norm) is plotted. In general the error decreases with the number of observations and the reconstruction is robust with noise. Finally, by observing the noiseless case (dashed curve) we see that when $n \geq 13$ the error becomes practically zero indicating fully compressed acquisition of the calcium traces with a roughly $4\times$ undersampling factor. We will see below that this undersampling factor is inversely proportional to the firing rate: we can recover highly sparse spike signals $S$ using very few measurements $n$.

## 2.2 Estimation of the spatial matrix $A$

The above algorithm assumes that the underlying neurons have known locations, i.e., the matrix $A$ is known. In some cases $A$ can be estimated a-priori by running a conventional raster-scanning experiment at a high spatial resolution and locating the active voxels. However this approach is expensive and can still be challenging due to noise and possible spatial overlap between different neurons. To estimate $A$ within the compressive framework we note that the baseline-subtracted spatiotemporal calcium matrix $F$ (see eqs. (2) and (4)) can be written as $\bar{F} = F - \boldsymbol{b}\mathbf{1}_T^T = AC$; thus $\text{rank}(\bar{F}) \leq N$ where $N$ is the number of underlying neurons, with typically $N \ll d$. Since $N$ is also in general unknown we estimate $\bar{F}$ by solving a nuclear norm penalized problem (Recht et al., 2010)

$$
\underset{\bar{F}}{\text{minimize}} \quad \sum_{t=1}^{T} \frac{1}{2}(\bar{\boldsymbol{y}}(t) - B_t \bar{\boldsymbol{f}}(t))^T \Sigma_t^{-1}(\bar{\boldsymbol{y}}(t) - B_t \bar{\boldsymbol{f}}(t)) - \log p(\bar{F}) + \lambda_{NN}\|\bar{F}\|_* \quad \text{(P-NN)}
$$

$$
\text{subject to} \quad \bar{F}G^T \geq 0, \bar{\boldsymbol{f}}(1) \geq 0,
$$

where $\| \cdot \|_*$ denotes the nuclear norm (NN) of a matrix (i.e., the sum of its singular values), which is a convex approximation to the nonconvex rank function (Fazel, 2002). The prior of $\bar{F}$ can be chosen in a similar fashion as $\log p(C)$, i..e, $\log p(\bar{F}) \propto -\boldsymbol{\lambda}_F^T \bar{F} G^T \mathbf{1}_T$, where $\boldsymbol{\lambda}_F \in \mathbb{R}^d$. Although more complex than (P-QP), (P-NN) is again convex and can be solved efficiently using e.g. the ADMM method of Boyd et al. (2011). From the solution of (P-NN) we can estimate $N$ by appropriately thresholding the singular values of the estimated $\bar{F}$.[3] Having $N$ we can then use appropriately constrained nonnegative matrix factorization (NMF) methods to alternately estimate $A$ and $C$. Note that during this NMF step the baseline vector $\boldsymbol{b}$ can also be estimated jointly with $A$. Since NMF methods are nonconvex, and thus prone to local optima, informative initialization is important. We can use the solution of (P-NN) to initialize the spatial component $A$ using clustering methods, similar to methods typically used in neuronal extracellular spike sorting (Lewicki, 1998). Details are given in the supplement (along with some discussion of the estimation of the other parameters in this problem); we refer to Pnevmatikakis et al. (2013) for full details.

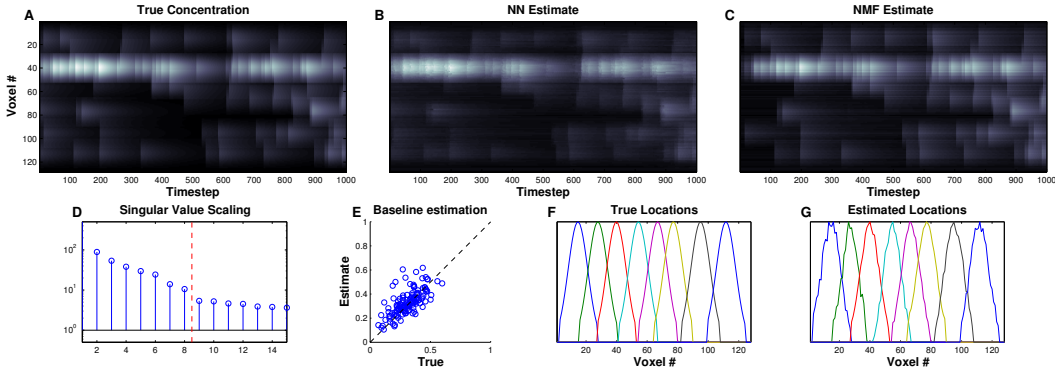

Figure 2: Estimating locations and calcium concentration from compressive calcium imaging measurements. A: True spatiotemporal concentration B: estimate by solving (P-NN) C: estimate by using NMF methods. D: Logarithmic plot of the first singular values of the solution of (P-NN), E: Estimation of baseline vector, F: true spatial locations G: estimated spatial locations. The NN-penalized method estimates the number of neurons and the NMF algorithm recovers the spatial and temporal components with high accuracy.

In Fig. 2 we present an application of this method to an example with $N = 8$ spatially overlapping neurons. For simplicity we consider neurons in a one-dimensional field with total number of voxels $d = 128$ and spatial positions shown in Fig. 2E. At each timestep we obtain just $n = 5$ noisy measurements using random projections on binary masks. From the solution to the NN-penalized problem (P-NN) (Fig. 2B) we threshold the singular values (Fig. 2D) and estimate the number of underlying neurons (note the logarithmic gap between the 8th and 9th largest singular values that enables this separation). We then use the NMF approach to obtain final estimates of the spatial locations (Fig. 2G), the baseline vector (Fig. 2E), and the full spatiotemporal concentration (Fig. 2C). The estimates match well with the true values. Note that $n < N \ll d$ showing that compressive imaging with significant undersampling factors is possible, even in the case of classical raster scanning protocol where the spatial locations are unknown.

## 3    Estimation of the phase transition curve in the noiseless case

The results presented above indicate that reconstruction of the spikes is possible even with significant undersampling. In this section we study this problem from a compressed sensing (CS) perspective in the idealized case where the measurements are noiseless. For simplicity, we also assume that $A = I$ (similar to a RAMP setup). Unlike the traditional CS setup, where a sparse signal (in some basis) is sensed with a dense fully supported random matrix, in our case the sensing matrix $B$ has a block-diagonal form. A standard justification of CS approaches proceeds by establishing that the sensing matrix satisfies the "restricted isometry property" (RIP) for certain classes of sparse signals

with high probability (w.h.p.); this property in turn guarantees the correct recovery of the parameters of interest (Candes and Tao, 2005). Yap et al. (2011) showed that for signals that are sparse in some orthogonal basis, the RIP holds for random block-diagonal matrices w.h.p. with a number of sufficient measurement that scales with the squared coherence between the sparse basis and the elementary (identity) basis. For non-orthogonal basis the RIP property has only been established for fully dense sensing matrices (Candes et al., 2011). For signals with sparse variations Ba et al. (2012) established perfect and stable recovery conditions under the assumption that the sensing matrix at each timestep satisfies certain RIPs, and the sparsity level at each timestep has known upper bounds.

While the RIP is a valuable tool for the study of convex relaxation approaches to compressed sensing problems, its estimates are usually up to a constant and can be relatively loose (Blanchard et al., 2011). An alternative viewpoint is offered from conic geometric arguments (Chandrasekaran et al., 2012; Amelunxen et al., 2013) that examine how many measurements are required such that the convex relaxed program will have a unique solution which coincides with the true sparse solution. We use this approach to study the theoretical properties of our proposed compressed calcium imaging framework in an idealized noiseless setting. When noise is absent, the quadratic program (P-QP) for the approximate MAP estimate converges to a linear program[4]:

$$\text{minimize}_C \ f(C), \ \text{subject to: } B\text{vec}(C) = \text{vec}(Y) \tag{P-LP}$$

with
$$f(C) = \begin{cases} (\boldsymbol{v} \otimes 1_N)^T \text{vec}(C), & (G \otimes I_d)\text{vec}(C) \geq 0 \\ \infty, & \text{otherwise} \end{cases}, \ \text{and } \boldsymbol{v} = G^T 1_T.$$

Here $\otimes$ denotes the Kronecker product and we used the identity $\text{vec}(CG^T) = (G \otimes I_d)\text{vec}(C)$. To examine the properties of (P-LP) we follow the approach of Amelunxen et al. (2013): For a fully dense sensing i.i.d. Gaussian (or random rotation) matrix $B$, the linear program (P-LP) will succeed w.h.p. to reconstruct the true solution $C_0$, if the total number of measurements $nT$ satisfies

$$nT \geq \delta(\mathcal{D}(f, C_0)) + O(\sqrt{TN}). \tag{5}$$

$\mathcal{D}(f, C_0)$ is the descent cone of $f$ at $C_0$, induced by the set of non-increasing directions from $C_0$, i.e.,

$$\mathcal{D}(f, C_0) = \cup_{\tau \geq 0} \left\{ \boldsymbol{y} \in \mathbb{R}^{N \times T} : f(C_0 + \tau \boldsymbol{y}) \leq f(C_0) \right\}, \tag{6}$$

and $\delta(\mathcal{C})$ is the "statistical dimension" (SD) of a convex cone $\mathcal{C} \subseteq \mathbb{R}^m$, defined as the expected squared length of a standard normal Gaussian vector projected onto the cone

$$\delta(\mathcal{C}) = \mathbb{E}_{\boldsymbol{g}} \|\Pi_{\mathcal{C}}(\boldsymbol{g})\|^2, \ \text{with } \boldsymbol{g} \sim \mathcal{N}(\boldsymbol{0}, I_m).$$

Eq. (5), and the analysis of Amelunxen et al. (2013), state that as $TN \to \infty$, the probability that (P-LP) will succeed to find the true solution undergoes a phase transition, and that the phase transition curve (PTC), i.e., the number of measurements required for perfect reconstruction normalized by the ambient dimension $NT$ (Donoho and Tanner, 2009), coincides with the normalized SD. In our case $B$ is a block-diagonal matrix (not a fully-dense Gaussian matrix), and the SD only provides an estimate of the PTC. However, as we show below, this estimate is tight in most cases of interest.

### 3.1 Computing the statistical dimension

Using a result from Amelunxen et al. (2013) the statistical dimension can also be expressed as the expected squared distance of a standard normal vector from the cone induced by the subdifferential (Rockafellar, 1970) $\partial f$ of $f$ at the true solution $C_0$:

$$\delta(\mathcal{D}(f, C_0) = \mathbb{E}_{\boldsymbol{g}} \inf_{\tau > 0} \min_{\boldsymbol{u} \in \tau \partial f(C_0)} \|\boldsymbol{g} - \boldsymbol{u}\|^2, \ \text{with } \boldsymbol{g} \sim \mathcal{N}(\boldsymbol{0}, I_{NT}). \tag{7}$$

Although in general (7) cannot be solved in closed form, it can be easily estimated numerically; in the supplementary material we show that the subdifferential $\partial f(C_0)$ takes the form of a convex polytope, i.e., an intersection of linear half spaces. As a result, the distance of any vector $\boldsymbol{g}$ from $\partial f(C_0)$ can be found by solving a simple quadratic program, and the statistical dimension can be estimated with a simple Monte-Carlo simulation (details are presented in the supplement). The characterization of (7) also explains the effect of the sparsity pattern on the SD. In the case where the sparse basis

is the identity then the cone induced by the subdifferential can be decomposed as the union of the respective subdifferential cones induced by each coordinate. It follows that the SD is invariant to coordinate permutations and depends only on the sparsity level, i.e., the number of nonzero elements. However, this result is in general not true for a nonorthogonal sparse basis, indicating that the precise location of the spikes (sparsity pattern) and not just their number has an effect on the SD. In our case the calcium signal is sparse in the non-orthogonal basis described by the matrix $G$ from (4).

## 3.2 Relation with the phase transition curve

In this section we examine the relation of the SD with the PTC for our compressive calcium imaging problem. Let $S$ denote the set of spikes, $\Omega = \mathrm{supp}(S)$, and $C$ the induced calcium traces $C = SG^{-T}$. As we argued, the statistical dimension of the descent cone $\mathcal{D}(f, C)$ depends both on the cardinality of the spike set $|\Omega|$ (sparsity level) and the location of the spikes (sparsity pattern). To examine the effects of the sparsity level and pattern we define the normalized expected statistical dimension (NESD) with respect to a certain distribution (e.g. Bernoulli) $\chi$ from which the spikes $S$ are drawn.

$$\tilde{\delta}(k/NT, \chi) = \mathbb{E}_{\Omega \sim \chi} \left[ \delta(\mathcal{D}(f, C))/NT \right], \quad \text{with } \mathrm{supp}(S) = \Omega, \text{ and } \mathbb{E}_{\Omega \sim \chi} |\Omega| = k.$$

In Fig. 3 we examine the relation of the NESD with the phase transition curve of the noiseless problem (P-LP). We consider a setup with $N = 40$ point neurons ($A = I_d, d = N$) observed over $T = 50$ timesteps and chose discrete time constant $\gamma = 0.99$. The spike-times of each neuron came from the same distribution and we considered two different distributions: (i) Bernoulli spiking, i.e., each neuron fires i.i.d. spikes with the probability $k/T$, and (ii) desynchronized periodic spiking where each neuron fires deterministically spikes with discrete frequency $k/T$ timesteps$^{-1}$, and each neuron has a random phase. We considered two forms of spikes: (i) with nonnegative values ($s_i(t) \geq 0$), and (ii) with binary values ($s_i(t) = \{0, 1\}$), and we also considered two forms of sensing matrices: (i) with time-varying matrix $B_t$, and (ii) with constant, fully supported matrices $B_1 = \ldots = B_T$. The entries of each $B_t$ are again drawn from an i.i.d. fair Bernoulli distribution. For each of these 8 different conditions we varied the expected number of spikes per neuron $k$ from 1 to $T$ and the number of observed measurements $n$ from 1 to $N$. Fig. 3 shows the empirical probability that the program (P-LP) will succeed in reconstructing the true solution averaged over a 100 repetitions. Success is declared when the reconstructed spike signal $\hat{S}$ satisfies[5] $\|\hat{S} - S\|_F / \|S\|_F < 10^{-3}$. We also plot the empirical PTC (purple dashed line), i.e., the empirical 50% success probability line, and the NESD (solid blue line), approximated with a Monte Carlo simulation using 200 samples, for each of the four distinct cases (note that the SD does not depend on the structure of the sensing matrix $B$).

In all cases, our problem undergoes a sharp phase transition as the number of measurements per timestep varies: in the white regions of Fig. 3, $S$ is recovered essentially perfectly, with a transition to a high probability of at least some errors in the black regions. Note that the phase transitions are defined as functions of the sparsity index $k/T$; the signal sparsity sets the compressibility of the data. In addition, in the case of time-varying $B_t$, the NESD provides a surprisingly good estimate of the PTC, especially in the binary case or when the spiking signal is actually sparse ($k/T < 0.5$), a result that justifies our overall approach. Although using time-varying sensing matrices $B_t$ leads to better results, compression is also possible with a constant $B$. This is an important result for implementation purposes where changing the sensing matrix might be a costly or slow operation. On a more technical side we also observe the following interesting properties:

- Periodic spiking requires more measurements for accurate deconvolution, a property again predicted by the SD. This comes from the fact that the sparse basis is not orthogonal and shows that for a fixed sparsity level $k/T$ the sparsity pattern also affects the number of required measurements. This difference depends on the time constant $\gamma$. As $\gamma \to 0$, $G \to I$; the problem becomes equivalent to a standard nonnegative CS problem, where the spike pattern is irrelevant.
- In the Bernoulli spiking nonnegative case, the SD is numerically very close to the PTC of the standard nonnegative CS problem (not shown here), adding to the growing body of evidence for universal behavior of convex relaxation approaches to CS (Donoho and Tanner, 2009).
- In the binary case the results exhibit a symmetry around the axis $k/T = 0.5$. In fact this symmetry becomes exact as $\gamma \to 1$. In the supplement we prove that this result is predicted by the SD.

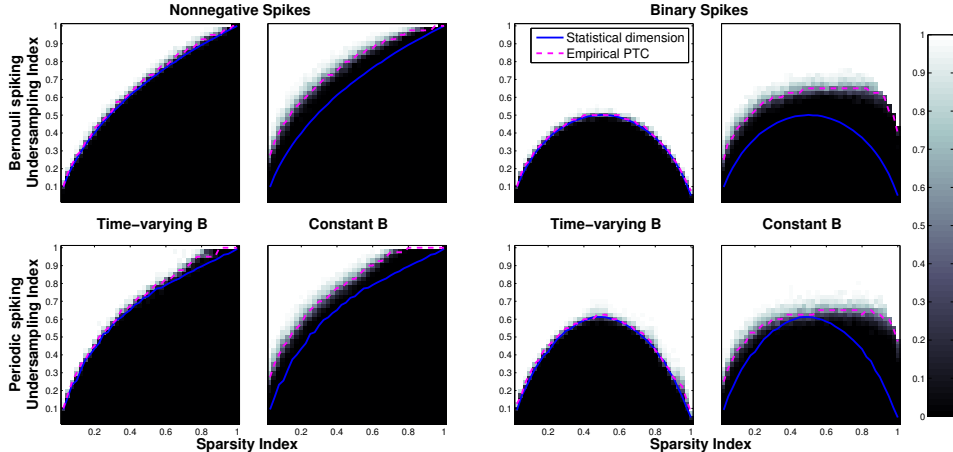

Figure 3: Relation of the statistical dimension with the phase transition curve for two different spiking distributions (Bernouli, periodic), two different spike values (nonnegative, binary), and two classes of sensing matrices (time-varying, constant). For each panel: x-axis normalized sparsity $k/T$, y-axis undersampling index $n/N$. Each panel shows the empirical success probability for each pair $(k/T, n/N)$, the empirical $50\%$-success line (dashed purple line) and the SD (blue solid line). When $B$ is time-varying the SD provides a good estimate of the empirical PTC.

As mentioned above, our analysis is only approximate since $B$ is block-diagonal and not fully dense. However, this approximation is tight in the time-varying case. Still, it is possible to construct adversarial counterexamples where the SD approach fails to provide a good estimate of the PTC. For example, if all neurons fire in a completely synchronized manner then the required number of measurements grows at a rate that is not predicted by (5). We present such an example in the supplement and note that more research is needed to understand such extreme cases.

## 4    Conclusion

We proposed a framework for compressive calcium imaging. Using convex relaxation tools from compressed sensing and low rank matrix factorization, we developed an efficient method for extracting neurons' spatial locations and the temporal locations of their spikes from a limited number of measurements, enabling the imaging of large neural populations at potentially much higher imaging rates than currently available. We also studied a noiseless version of our problem from a compressed sensing point of view using newly introduced tools involving the statistical dimension of convex cones. Our analysis can in certain cases capture the number of measurements needed for perfect deconvolution, and helps explain the effects of different spike patterns on reconstruction performance.

Our approach suggests potential improvements over the standard raster scanning protocol (unknown locations) as well as the more efficient RAMP protocol (known locations). However our analysis is idealistic and neglects several issues that can arise in practice. The results of Fig. 1 suggest a tradeoff between effective compression and SNR level. In the compressive framework the cycle length can be relaxed more easily due to the parallel nature of the imaging (each location is targeted during the whole "cycle"). The summed activity is then collected by the photomultiplier tube that introduces the noise. While the nature of this addition has to be examined in practice, we expect that the observed SNR will allow for significant compression. Another important issue is motion correction for brain movement, especially in vivo conditions. While new approaches have to be derived for this problem, the novel approach of Cotton et al. (2013) could be adaptable to our setting. We hope that our work will inspire experimentalists to leverage the proposed advanced signal processing methods to develop more efficient imaging protocols.

#### Acknowledgements

LP is supported from an NSF career award. This work is also supported by ARO MURI W911NF-12-1-0594.

## Footnotes

[1]Generalization to general $AR(p)$ processes is straightforward, but we keep $p = 1$ for simplicity.

[2]This choice is merely for simplicity; more general prior distributions can be incorporated in our framework.

[3]To reduce potential shrinkage but promote low-rank solutions this "global" NN penalty can be replaced by a series of "local" NN penalties on spatially overlapping patches.

[4]To illustrate the generality of our approach we allow for arbitrary nonnegative spike values in this analysis, but we also discuss the binary case that is of direct interest to our compressive calcium framework.

[5]When calculating this error we excluded the last 10 timesteps. As every spike is filtered by the AR process it has an effect for multiple timelags in the future and an optimal encoder has to sense it over multiple timelags. This number depends only on $\gamma$ and not on the length $T$, and thus this behavior becomes negligible as $T \to \infty$.

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
