[Supplementary Material · supplement.pdf]


# Supplementary material
# Sparse nonnegative deconvolution for compressive calcium imaging: algorithms and phase transitions

**Eftychios A. Pnevmatikakis and Liam Paninski**
Columbia University

## A  Parallel first order implementation using the ADMM method

The quadratic problem (P-QP) for finding the MAP estimate (repeated here for simplicity)

$$\underset{C}{\text{minimize}} \quad \sum_{t=1}^{T} \frac{1}{2} (\bar{\boldsymbol{y}}(t) - B_t \boldsymbol{c}(t))^T \Sigma_t^{-1} (\bar{\boldsymbol{y}}(t) - B_t \boldsymbol{c}(t)) + \boldsymbol{\lambda}^T C G^T \mathbf{1}_T \qquad \text{(P-QP)}$$

$$\text{subject to} \quad CG^T \geq 0, \boldsymbol{c}(1) \geq 0,$$

can be also expressed as

$$\underset{C,Z}{\text{minimize}} \quad \sum_{t=1}^{T} \frac{1}{2} (\bar{\boldsymbol{y}}(t) - B_t \boldsymbol{c}(t))^T \Sigma_t^{-1} (\bar{\boldsymbol{y}}(t) - B_t \boldsymbol{c}(t)) + \boldsymbol{\lambda}^T Z G^T \mathbf{1}_T$$

$$\text{subject to} \quad C = Z, CZ^T \geq 0, \boldsymbol{z}(1) \geq 0.$$

Using the alternate direction method of multipliers (ADMM) (Boyd et al., 2011) we solve the problem using the following iterative scheme until convergence:

$$C^{k+1} = \arg\min_{C} \sum_{t=1}^{T} \frac{1}{2} (\bar{\boldsymbol{y}}(t) - B_t \boldsymbol{c}(t))^T \Sigma_t^{-1} (\bar{\boldsymbol{y}}(t) - B_t \boldsymbol{c}(t)) + (\rho/2)\|C - Z^k + U^k\|^2$$

$$Z^{k+1} = \arg\min_{Z:ZG^T \geq 0, \boldsymbol{z}(1) \geq 0} \boldsymbol{\lambda}^T Z G^T \mathbf{1}_T + (\rho/2)\|C^{k+1} - Z + U^k\|^2$$

$$U^{k+1} = U^k + C^{k+1} - Z^{k+1},$$

with $\rho > 0$. Now for $C^{k+1}$ each $\boldsymbol{c}(t)$ can be estimated in parallel by solving a simple unconstrained quadratic program. The Hessian of each of these programs is of the form $H_t = \rho I_d + B_t^T \Sigma_t^{-1} B_t$ and therefore can be inverted in $O(Nn^2)$ time via the application of the Woodbury matrix identity (remember that each $B_t \in \mathbb{R}^{n \times d}$). Similarly, the estimation of $Z^{k+1}$ can be split into $N$ parallel programs each of which determines a row of $Z^{k+1}$ with cost $O(T)$ via a plain log-barrier method, similar to Vogelstein et al. (2010). Unlike the interior point method described in Sec. 2.1, the ADMM method is a first order method and typically requires more iterations for convergence. However, it may be a method of choice in the case where the spatial dimensionality is high and parallel computing resources are available.

# B   Parameter estimation details

## B.1   Choice of number of neurons

To estimate the number of neurons we compute the vector $\boldsymbol{\sigma}$ of the singular values of $\hat{F}$, where $\hat{F}$ is the solution of the NN-penalized problem (P-NN). We then construct the vector of consecutive ratios $\boldsymbol{a}$ with $a(i) = \sigma(i)/\sigma(i+1)$ and find the local minima of $\boldsymbol{a}$. We pick $N$ as the location of the first local minimum of $\boldsymbol{a}$ such that the $N$ chosen singular values capture a large fraction (e.g. 99%) of $\|\hat{F}\|_F^2$. We have observed that this method in general works well in practice.

## B.2   NMF step

We first note that the baseline vector $\boldsymbol{b}$ can be jointly estimated with the matrix $A$, since it can be incorporated as an additional column of $A$ that is multiplied with an additional row of $C$ where each entry is equal to 1, i.e., the spatiotemporal calcium matrix can be written as

$$F = [A, \boldsymbol{b}] \begin{bmatrix} C \\ \mathbf{1}_T^T \end{bmatrix}.$$

Given $A$ and $\boldsymbol{b}$, the calcium traces $C$ for all the neurons can be estimated by solving the (P-QP) problem presented in Sec. 2. Given $C$, the log-likelihood of $[A, \boldsymbol{b}]$ can be expressed as

$$\log(Y|A, \boldsymbol{b}; C) \propto -\frac{1}{2}\|\Sigma^{-1/2}B((C^T \otimes I_d)\mathrm{vec}(A) + (\mathbf{1}_T \otimes I_d)\boldsymbol{b} - \mathrm{vec}(Y))\|^2. \tag{S-1}$$

To jointly estimate $A$ and $\boldsymbol{b}$ we can maximize (S-1) subject to nonnegativity constrains $A \geq 0$, $\boldsymbol{b} \geq \mathbf{0}$. Additional regularizers can be introduced to $A$ to penalize the size of each neuron (via an $l_1$-norm on $A$) or to smooth the shape of each neuron (via a spatial derivative penalty on each row of $A$). Note that since $A \geq 0$ and $\boldsymbol{b} \geq 0$, the $l_1$-norm just becomes the sum of the elements and therefore the problem remains a nonnegative least squares problem that can be solved efficiently (e.g. with a log-barrier method).

   To initialize the NMF procedure we start from the solution of the NN-penalized problem (P-NN) and initialize the spatial component as follows:

1. Extract the "voxel" spikes from the solution of problem (P-NN)

$$\hat{S} = \hat{F}G^T$$

2. Threshold the extracted spikes with a relatively high threshold (e.g. 90% quantile).
3. Perform $k$-means clustering of the spike vectors at each timestep, with $N+1$ clusters and discard the cluster with the centroid closest to zero.
4. Use the remaining clusters to initialize $A^0$.
5. The baseline vector can be initialized by computing the remainder between the solution $\hat{F}$ and the extracted spikes and taking the mean for each voxel.

A full presentation of these methods will be pursued elsewhere (Pnevmatikakis et al., 2013).

# C   Computing the statistical dimension

We first rewrite the function $f$ for convenience:

$$f(C) = \begin{cases} (\boldsymbol{v} \otimes 1_N)^T \mathrm{vec}(C), & (G \otimes I_d)\mathrm{vec}(C) \geq 0 \\ \infty, & \text{otherwise} \end{cases} \text{, and } \boldsymbol{v} = G^T \mathbf{1}_T.$$

We can express $\delta(\mathcal{D}(f,C))$ in terms of the SDs of the descent cones induced by the rows of $C$. Let $\boldsymbol{c}_i$ denote the $i$-th row of $C$ (in column format) and define the function $f_r(\boldsymbol{c}) = \boldsymbol{v}^T\boldsymbol{c}$, for $G\boldsymbol{c} \geq \boldsymbol{0}$ (and $f_r(\boldsymbol{c}) = \infty$ otherwise). Note that $f(C) = \sum_{i=1}^N f_r(\boldsymbol{c}_i)$. Now if $\mathcal{D}(f_r, \boldsymbol{c}_i)$ is the descent cone of $f_r$ at $\boldsymbol{c}_i$, since the sparsifying basis does not multiplex the spikes of the different neurons we have

$$\mathcal{D}(f,C) = \mathcal{D}(f_r, \boldsymbol{c}_1) \times \mathcal{D}(f_r, \boldsymbol{c}_2) \times \ldots \times \mathcal{D}(f_r, \boldsymbol{c}_N) \implies \delta(\mathcal{D}(f,C)) = \sum_{i=1}^N \delta(\mathcal{D}(f_r, \boldsymbol{c}_i)), \qquad \text{(S-2)}$$

where the last equality follows from a property of the SD of direct product cones (Amelunxen et al., 2013). Using the expression of the statistical dimension in terms of the subdifferential (7), we have

$$\delta(\mathcal{D}(f_r, \boldsymbol{c})) = \mathbb{E}_{\boldsymbol{g}} \inf_{\tau > 0} \min_{\boldsymbol{u} \in \tau \partial f_r(\boldsymbol{c})} \|\boldsymbol{g} - \boldsymbol{u}\|^2.$$

To characterize the subdifferential consider the function $z : \mathbb{R}^T \mapsto \mathbb{R}$ with $z(\boldsymbol{x}) = \boldsymbol{1}_T^T\boldsymbol{x}$ if $\boldsymbol{x} \geq \boldsymbol{0}$ (pointwise), and $z(\boldsymbol{x}) = \infty$ otherwise. Then if $\Omega$ is the set of entries where $\boldsymbol{x}$ is nonzero, and $\Omega^c$ its complement, we have

$$\partial z(\boldsymbol{x}) = \boldsymbol{w}, \quad \text{with} \begin{cases} w(j) = 1, & j \in \Omega, \\ w(j) \leq 1, & j \in \Omega^c \end{cases}.$$

For our case note that $f_r(\boldsymbol{c}) = z(G\boldsymbol{c})$. Let $\Omega_i$ the set of spiketimes of neuron $i$ and $\Omega_i^c$ its complement in the set $[1, \ldots, T]$. Using the relation for affine transformations of the subdifferential $\partial f_r(\boldsymbol{c}) = G^T \partial z(G\boldsymbol{c})$, the subdifferential at $\boldsymbol{c}_i$, $\partial f_r(\boldsymbol{c}_i)$, can be characterized as

$$\partial f_r(\boldsymbol{c}_i) = G^T\boldsymbol{w}, \text{ with } \begin{cases} w(j) = 1, \ j \in \Omega_i \\ w(j) \leq 1, \ j \in \Omega_i^c \end{cases} \qquad \text{(S-3)}$$

Using (7) the SD equals the average value over all standard normal i.i.d. vectors $\boldsymbol{g}$ of the quadratic programs

$$\underset{\boldsymbol{w}, \tau}{\text{minimize}} \ \|\boldsymbol{g} - G^T\boldsymbol{w}\|^2, \quad \text{subject to: } \tau \geq 0, \{w(j) = \tau, j \in \Omega_i\}, \{w(j) \leq \tau, j \in \Omega_i^c\}. \qquad \text{(SD-QP)}$$

The statistical dimension can be estimated with a simple Monte Carlo simulation by averaging the values of a series of quadratic programs. Since $G$ is a bidiagonal matrix, each of the quadratic programs can be solved in $O(T)$ time using a standard interior point method. Note that the sparsity pattern here matters as opposed to the function $z(\boldsymbol{x})$ where the expected distance from the subdifferential cone is invariant under permutations.

# D  Symmetry of the statistical dimension in the binary case

The above analysis assumes that the spikes signal $S$ takes arbitrary nonnegative values. In the case where the spikes are restricted to take only binary $\{0,1\}$ values, a similar analysis can be carried: By inserting the additional condition $G\boldsymbol{c} \leq 1$, i.e., $f_r(\boldsymbol{c}) = \infty$ if $G\boldsymbol{c}_1 \not\leq 1$, the subdifferential $\partial f_r(\boldsymbol{c}_i)$ is now given by

$$\partial f_r(\boldsymbol{c}_i) = G^T\boldsymbol{w}, \text{ with } \begin{cases} w(j) \geq 1, \ j \in \Omega_i \\ w(j) \leq 1, \ j \in \Omega_i^c \end{cases}. \qquad \text{(S-4)}$$

Note that (S-4) is very similar to (S-3) with the difference that $w(j) \geq 1$ for $j \in \Omega_i$.

When $\gamma = 1$ the objective function $f_r$ becomes a nonnegative total-variation norm. We now prove the statistical dimension with is asymptotically symmetric. Let two binary spiking signals $s_1, s_2$ that satisfy $s_1 = 1 - s_2$ and also let $\Omega_1, \Omega_2$ the corresponding sets of spiketimes, and $c_1, c_2$ the corresponding calcium traces. It holds that $\Omega_1 = [1, \ldots, T] \backslash \Omega_2$. Define $\mathcal{C}_1, \mathcal{C}_2$ the set of conic constraints that these signals induce when computing the statistical dimension of the descent cones (eq. (SD-QP)):

$$\mathcal{C}_1 = \{\boldsymbol{w}, \tau : (\tau \geq 0) \cap (w(j) \geq \tau, j \in \Omega_1) \cap (w(j) \leq \tau, j \in \Omega_2)\}$$
$$\mathcal{C}_2 = \{\boldsymbol{w}, \tau : (\tau \geq 0) \cap (w(j) \geq \tau, j \in \Omega_2) \cap (w(j) \leq \tau, j \in \Omega_1)\}$$

Now define the functions $h_i : \mathbb{R}^T \mapsto \mathbb{R}_+$ $(i = 1, 2)$ as the value of the quadratic problems

$$h_i(\boldsymbol{g}) = \min_{\boldsymbol{w}, \tau \in \mathcal{C}_i} \|\boldsymbol{g} - G^T \boldsymbol{w}\|^2. \tag{S-5}$$

By making the change of variables $\boldsymbol{w} \leftarrow \boldsymbol{w} - \tau$, the set of constraints and the value functions become respectively

$$\tilde{\mathcal{C}}_1 = \{\boldsymbol{w}, \tau : (\tau \geq 0) \cap (w(j) \geq 0, j \in \Omega_1) \cap (w(j) \leq 0, j \in \Omega_2)\}$$
$$\tilde{\mathcal{C}}_2 = \{\boldsymbol{w}, \tau : (\tau \geq 0) \cap (w(j) \geq 0, j \in \Omega_2) \cap (w(j) \leq 0, j \in \Omega_1)\},$$

and the value functions of (S-5) can be expressed as

$$h_i(\boldsymbol{g}) = \min_{\boldsymbol{w}, \tau \in \tilde{\mathcal{C}}_i} \|\boldsymbol{g} - G^T \boldsymbol{w} + \tau G^T \mathbf{1}_T\|^2.$$

If $\gamma = 1$ we have $G^T \mathbf{1}_T = [0, 0, \ldots, 0, 1]^T$ and therefore we argue that the contribution of the term $\tau G^T \mathbf{1}_T$ in the value function is of order $O(1)$ and that we have

$$h_i(\boldsymbol{g}) = \bar{h}_i(\boldsymbol{g}) + O(1),$$

where $\bar{h}_i$ is defined as

$$\bar{h}_i(\boldsymbol{g}) = \min_{\boldsymbol{w} \in \bar{\mathcal{C}}_i} \|\boldsymbol{g} - G^T \boldsymbol{w}\|^2,$$

with

$$\bar{\mathcal{C}}_1 = \{\boldsymbol{w} : (w(j) \geq 0, j \in \Omega_1) \cap (w(j) \leq 0, j \in \Omega_2)\}$$
$$\bar{\mathcal{C}}_2 = \{\boldsymbol{w} : (w(j) \geq 0, j \in \Omega_2) \cap (w(j) \leq 0, j \in \Omega_1)\}$$

Now note that $\bar{\mathcal{C}}_1 = -\bar{\mathcal{C}}_2$ and therefore $\bar{h}_1(\boldsymbol{g}) = \bar{h}_2(-\boldsymbol{g})$. Therefore

$$\delta(\mathcal{D}(f_r, \boldsymbol{c}_1)) = \mathbb{E}_{\boldsymbol{g}}(h_1(\boldsymbol{g})) = \mathbb{E}_{\boldsymbol{g}}(\bar{h}_1(\boldsymbol{g})) + O(1) = \mathbb{E}_{\boldsymbol{g}}(\bar{h}_2(\boldsymbol{g})) + O(1) = \delta(\mathcal{D}(f_r, \boldsymbol{c}_2)) + O(1)$$

and in the asymptotic regime

$$\lim_{T \to \infty} \frac{1}{T} \delta(\mathcal{D}(f_r, \boldsymbol{c}_1)) = \lim_{T \to \infty} \frac{1}{T} \delta(\mathcal{D}(f_r, \boldsymbol{c}_2)),$$

which establishes our symmetry argument.

# E Counterexamples

As the mentioned in the main text, the condition (5)

$$nT \geq \delta(\mathcal{D}(f, C_0)) + O(\sqrt{NT}), \tag{S-6}$$

is exact only when $B$ is a fully dense i.i.d. random matrix. Here we construct a counterexample where the statistical dimension fails to provide a tight lower bound on the phase transition curve. Consider again $N$ neurons. Each neuron has Bernoulli spiking with probability of spike per timestep $k/T$, but now assume that all the neurons are perfectly synchronized. From (S-2), the statistical dimension $\delta(\mathcal{D}(f, C))$ can be decomposed as

$$\delta(\mathcal{D}(f, C)) = \sum_{i=1}^{N} \delta(\mathcal{D}(f_r, \boldsymbol{c}_i)),$$

i.e., it does not depend on the spike correlations between the different neurons. However we expect that the synchrony between the neurons will make the reconstruction harder. For example, when the sensing matrix $B$ is constant and the spikes have nonnegative values, then compressive acquisition is impossible since the dense calcium signal will always projected in the same lower-dimensional subspace spanned by the constant $B$, and thus it cannot be recovered.

Figure S-1: Relation of the statistical dimension with the phase transition curve in a synchronized firing case. For each panel: x-axis normalized sparsity $k/T$, y-axis undersampling index $n/N$. Left: Nonnegative spikes. Right: Binary spikes. The statistical dimension (solid blue line) does not provide a good estimate of the empirical PTC (purple dashed line) for this case.

In Fig. S-1 we consider a time-varying sensing matrix $B$ and examine the cases where the spikes take nonnegative values (left) and binary values (right). We again considered $N = 40$ and $T = 50$ and removed the last 10 timesteps from the calculation of the reconstruction error. We again varied the probability of spiking $k/T$ from 0 to 1 and the number of measurements per timestep $n$ from 1 to $N$. In both cases we see that the statistical dimension fails to provide a tight bound on the empirical PTC. However, note that compression is still possible for the nonnegative spikes case, if the signal is very sparse, a result which indicates that occasional synchronies can be tolerated

within the proposed compressive framework. In the binary case, compression is always possible, a result that is expected from the relatively simple structure of binary signals, and is important for our compressive calcium imaging framework, where the spiking signal is expect to be binary (present or absent).

Note that the failure of the SD to capture the PTC should not be attributed to the decomposition property (S-2) which ignores the correlations between the different neurons, but rather on the block-diagonal structure of the sensing matrix $B$. If $B$ is fully dense, then the SD coincides with the PTC as is predicted in general result of Amelunxen et al. (2013) (data not shown). More research is needed to understand the effects of restricting $B$ to be a block-diagonal matrix.