[Reviews · NeurIPS 2013]

Submitted by Assigned_Reviewer_4

This paper proposes a method for compressive sensing calcium imaging, which takes advantage of the sparseness of neuronal activity. The method reduces the number of measurements significantly and uses recently developed mathematical techniques to decode these measurements. Furthermore, the number of measurements necessary for full recovery is studied using cutting-edge mathematical techniques.

The strength of the paper is in the application of fashionable mathematical techniques to address an important neuroscience problem. The weakness of the paper is that no analysis of actual data is presented. I realize that the required datasets may not yet exist. However, expecting that an experimentalist would be able to read this paper and develop a compressive sensing imaging protocol seems a bit unrealistic to me. I think that the key idea could be communicated on a much simpler level than that of the current presentation.
Summary: Modern treatment of compressive sensing calcium imaging

Submitted by Assigned_Reviewer_5

2-photon calcium population imaging has made it possible to record from large populations of neurons, but given that only one voxel can be imaged at any time, there are still inevitable trade-offs between temporal resolution and population size. This paper proposes to use techniques from compressive sensing (which, as the authors describe, have by now been utilized in a variety of other imaging settings, including confocal fluorescence imaging) for neural population imaging-- this approach promises to use voxel measurements more effectively, and thus opens the possibility of recording from much larger populations.

The authors set up a simple linear model of the measurement process with fixed baseline, AR(1) temporal dependence and binary spikes, and spell out both a 'standard' reconstruction algorithm for it and also one which simultaneously estimates the locations of neurons, and analyze both algorithms using simulation and theory (only for the 'standard' approach).')


Detailed comments:
a) The paper seems a bit disjointed jumping between the 'known locations' and 'unknown locations' cases-- I think concentrating on the 'known locations' case and using the additional space to provide more details would have made for a stronger and more readable paper.
b) The weights are +-1 in the imaging matrix B-- is it possible to image 'negatively', and if not, how does this affect the applicability of the approach?
c) Neural activity is sparse but (might be) synchronized in time-- how would this affect the applicability of the method?


Quality: While there are open questions as to whether and how well this works on real data, the analysis and methods provided are sound (although I am in no position to comment on the rigour of the theory). The theory seems a bit disjoint from the simulation results.
Clarity: This is a dense paper, and there are not really enough details to understand the theory. It would probably have made for a stronger to concentrate on a few points and work them out more clearly.
Originality: While CS imaging has been performed in other imaging domains, to my knowledge, this is the first application to 2 photon calcium imaging.
Significance: This is a conceptual and even speculative paper-- however, if this method is really successfully put into practice, the impact could be dramatic.

Summary: This paper proposes a new algorithmic framework for calcium imaging based on randomized measurements, and shows hat such an approach has potential to allow neural population measurements to be scaled to larger population sizes than previously possible. While the paper seems a bit `conceptual' and there are open questions regarding the applibility of this approach to real data, the idea is very interesting and potentially powerful, and therefore a great paper for NIPS.



Submitted by Assigned_Reviewer_6

[EDIT] I read the rebuttal and removed point 3 from my review since it has been sufficiently addressed. Although I'm still not 100% happy with the responses to points 1 and 2 I upped the score to 8.


The authors propose a framework for two-photon imaging based on compressed sensing. In the first half of the paper they use simulations to demonstrate the feasibility of estimating calcium traces, non-negative firing rates (closely corresponding to spike times in a high-SNR situation) and spatial intensity profiles of the imaged neurons based on random projections of the data. In the second part they show that in the noiseless case the quality of reconstruction undergoes a phase transition. They derive an approximation to the phase transition curve, which they show to be quite accurate under some conditions.

I am generally enthusiastic about the approach since it could have a lot of potential to substantially improve the efficiency of two-photon imaging experiments, which generally suffer from the inherent trade-off between data quality and number of neurons that can be imaged. Although I am unable to verify the math, in particular in section 3, judging from the successful verification through simulations I believe it's correct.

Besides the stylistic complaint that the paper is written in a rather technical way with a lot of jargon, my major concern is that the authors did not do a good job at convincing the reader that the approach will actually be feasible or even useful in practice. There are three main issues related to this concern:

1. Since the method is sensitive to noise (Fig. 1) it is important to test it with realistic noise levels. In practice, noise levels are very high with calcium imaging due to shot noise, in particular so if one images fast as the authors propose. Although I don't know for sure what a typical SNR (as defined in line 189) would be for a real experiment, my guess would be that it's closer to 0 dB than to 20 dB as used in the Fig. 1 and I wouldn't be surprised if it's even substantially lower than 0 dB. Since this is a very fundamental problem, the authors should present some convincing evidence that the SNRs they use are realistic.

2. For estimating the locations, it looks like although the neurons were overlapping there was no background (neuropil) activity between the neurons and the neurons were actually covering the major fraction of the imaged space. However, in practice the situation is dramatically different. In a 3d volume cell bodies make up at most 10% (probably less) of the voxels while the remainder consists of neuropil (axons and dendrites), which display calcium activity as well (although usually highly correlated and with faster dynamics). Although I would guess that the result would be a few (large) singular values that could be excluded based on their spatial profile, I cannot judge whether the algorithm (P-NN) is really not affected in a major way by this. In addition, using SVD to identify neurons assumes they fire uncorrelated, which is unlikely to be true for most experiments. Since one of the main goals of imaging many neurons at the same time is to characterize their joint activity, this could be a major caveat and should at least be discussed.

[3. removed since clarified by rebuttal]

The above concerns would be somewhat less of a problem if the goal of the paper was to just lay out the basic mathematical framework, but then the authors should be more upfront about the various practical problems that may arise. Currently, the manuscripts reads somewhat like the method could be used to build an actual imaging setup, but I think we are still relatively far away from that goal.
Summary: Potentially very interesting approach to calcium imaging based on compressed sensing, but the practical feasibility is somewhat questionable and could be better addressed.
Author Feedback

Author rebuttal: We thank the reviewers for their constructive comments. There are 2 major concerns: a) The paper is rather dense and technical. b) It is unclear whether our proposed framework would be feasible and gainful in practice.

Our main goal is to propose the compressive calcium imaging framework and efficient algorithms for dealing with such obtained data. However, we also strongly believe that a theoretical analysis is crucial. We provide quantitative estimates on the performance gains that clearly show the potential of the proposed methods. We also provide novel qualitative results on compressed sensing with a non-orthogonal basis that, although not our main focus, can be valuable for the general NIPS audience. Many details appear in the appendix to make the paper more self-contained.

It's true that the ultimate test for this framework would be in practice. An implementation is feasible since similar compressive imaging frameworks have been implemented successfully (Studer et al. 2012), and our algorithms are applicable since they build on their non-compressive counterparts (B=I) which have already been proven successful to analyzing classical raster-scanning and random access calcium imaging data. We can revise the paper in favor of a more clear presentation of the general idea and the implementation challenges but getting into specific details would be more speculative and undesirable.

Answers to specific concerns:
Assigned_reviewer_4
While we appreciate the effort, we found this review (compared to the others) somewhat less constructive and a bit dismissive. We would appreciate if the reviewer would make his/her concerns more explicit.

Assigned_reviewer_5
Known/Unknown locations: We think that treating the unknown locations case is practically important since their prior determination can be a hard and computationally expensive problem (see also below and lines 211-214).

Weights: This concern is valid: negative combinations cannot be straightforwardly implemented. However, what is important is that each measurement vector is sufficiently uncorrelated with the underlying neural activity. Our simulations show that changing the +-1 to a {0,1} matrix doesn't affect the performance of our methods. We'll revise the paper accordingly.

Synchronized activity: There are two aspects to this issue: First, synchronized neurons share the same temporal activity, reducing the rank of the spatiotemporal matrix and thus the degrees of freedom overall. This helps the applicability of our method. On the other hand, we noticed and described in the appendix that our theoretical analysis (which assumes known locations and a fully supported overall sensing matrix B, rather than the block-diagonal we have here) becomes less accurate in the case of massive synchronization. However, significant compression is still possible, and our analysis ignores the rank deficiency. So we believe that synchronization will not affect the qualitative behavior of our method, although tight theoretical results are harder to obtain.

Assigned_reviewer_6
SNR: As the reviewer implies the SNR depends on the time spent at every location. In raster scanning approaches this time equals the duration of each imaging cycle divided by the total number of imaged pixels/voxels, giving an effective SNR ~ 5dB. In the compressive framework the cycle length can be relaxed more easily due to the parallel nature of the imaging (each location is targeted during the whole "cycle"). The summed activity is then collected by the photomultiplier tube that introduces the noise. So while the nature of this addition has to be examined in practice, we actually expect similar or better imaging quality while at the same time the overall imaging rate remains significantly higher. A simulation comparing the noise sensitivity of our framework to that of standard approaches could be helpful, but is omitted due to space constraints. Finally, we'd like to stress that the design of more efficient fluorescent calcium sensors is an active research area (GCaMP6 was just introduced) and the SNR keeps increasing.

Neuropil imaging: This fraction depends on the spatial resolution and preparation used in each experiment, and cases where most of the field is covered by cell bodies are common. Note that our approach can also be used for imaging dendritic/axonal activity as well which is also of interest, albeit harder due to the different nature of the calcium dynamics.

Correlations: Small/mid-level correlations don’t affect the number of significant singular values since they cannot be explained by a subspace with dimension smaller than the number of underlying neurons. The actual locations and temporal activity are then extracted with the matrix factorization step outlined in lines 228-231 and explained in greater detail in the appendix. The reviewer's concern arises when the neurons are (close to) perfectly synchronized. Then the algorithm would promote this synchronized activity, a result we see more as a feature than a bug (see also response to reviewer5)

Gains when locations are known: It is true that this information can be used to avoid redundancy by restricting imaging to these locations either through random access or in parallel (Nikolenko et al. 2008). However, this knowledge requires an initial step of imaging at a very high spatial resolution and can also be a hard problem in the case of overlapping neurons. More importantly, spatial sparsity is only the one aspect of redundancy, the other being the typically sparse neuron firing (lines 49-51). Fig 1 and the theory of section 3, indicate that when the exact locations are known (point neurons), our compressive framework can lead to substantial rate gains. Finally, random projections in our framework are implemented in parallel with a micromirror device (lines 126-128), and not through random access microscopy.

All of these valid concerns can be addressed in the revision.